# Cyclin-Dependent Kinase Inhibitor 2A/B Homozygous Deletion Prediction and Survival Analysis

**DOI:** 10.3390/brainsci13040548

**Published:** 2023-03-25

**Authors:** Jing Yang, Lei Li, Tao Luo, Chengsong Nie, Rui Fan, Deqiang Li, Rui Yang, Changru Zhou, Qian Li, Xiaofei Hu, Wei Chen

**Affiliations:** 1Department of Radiology, Southwest Hospital, Third Military Medical University (Army Medical University), Chongqing 400038, China; 2Institute of Pathology and Southwest Cancer Center, Southwest Hospital, Third Military Medical University (Army Medical University) and Key Laboratory of Tumor Immunopathology, Ministry of Education of China, Chongqing 400038, China; 3Department of Nuclear Medicine, Southwest Hospital, Third Military Medical University (Army Medical University), Chongqing 400038, China

**Keywords:** glioma, *CDKN2A/B* homozygous deletion, radiomics, survival analysis, MRI

## Abstract

Cyclin-Dependent Kinase Inhibitor 2A/B (*CDKN2A/B*) homozygous deletion was a significant prognostic factor for gliomas and affected the treatment strategy. However, the radiomic features of *CDKN2A/B* homozygous deletion in gliomas have not been developed, and whether the radiomic features and molecular subgroups can provide prognostic value in low-grade gliomas (LGGs) has yet to be studied. Thus, this study aimed to develop a predictive model of *CDKN2A/B* in gliomas and investigate the prognostic value of this biomarker and radiomic features in isocitrate dehydrogenase (*IDH*)-mutant LGGs. First, we developed the predictive model of *CDKN2A/B* homozygous deletion in 292 patients. The results revealed that radiomic features predict *CDKN2A/B* homozygous deletion with high accuracy and reliability. Subsequently, the prognostic survival models of 104 patients (*IDH*-mutant LGGs) were established, which provided an essential value for prognostic evaluation and indicated that *CDKN2A/B* homozygous deletion can be used as an independent predictor of prognosis in LGGs.

## 1. Introduction

Diffuse low- and intermediate-grade gliomas are among the most common infiltrative primary brain tumors [1,2]. The prognosis of low-grade gliomas (LGGs) varies greatly, especially in the isocitrate dehydrogenase (*IDH*)-mutant and *IDH*-wild-type gliomas [3]. Recent studies have reported that *IDH*-mutant gliomas typically present with lower histological grades and remain inert for many years; however, some transition to higher grades with worse clinical behavior later in the natural history of the disease, leading to treatment failure and poorer prognosis [4]. As such, traditional *IDH* classification methods are unable to aid in the development of personalized treatment plans to meet the challenges of dealing with gliomas.

*CDKN2A* is a gene located on chromosome 9, band p21.3 [5]. It is a negative regulator of cyclin-dependent kinases (*CDKs*) 4 and 6, which control the G1 to S cell cycle checkpoint by regulating the phosphorylation of the retinoblastoma protein family [4,6,7,8]. *CDKN2A* homozygous deletion leads to pathway activation and promotes proliferation. It has been identified as a target of therapeutic intervention [9,10].

Many studies have found that homozygous deletion of cyclin-dependent kinase inhibitor 2A/B (*CDKN2A/B*) in patients with *IDH*-mutant diffuse astrocytomas is a sign of poor prognosis [11,12]. *CDKN2A/B* homozygous deletion strongly correlates with shorter survival times and is an independent marker for World Health Organization (WHO) grade II and III *IDH*-mutant astrogliomas [12]. Subsequently, studies have also identified *CDKN2A/B* homozygous deletion as a significant prognostic marker of glioblastoma multiforme (GBM) [13]. Patients with *CDKN2A/B* homozygous deletion have a worse prognosis than those with other GBMs without homozygous deletion. Hence, *CDKN2A/B* homozygous deletion has been highlighted as a prognostic marker of astrocytoma mutants in the updated glioma classification for 2021 [14]. Detecting this deletion in patients is critical for appropriate treatment and prognosis.

Analysis of a genome sequence is the primary method for detecting *CDKN2A* deletion. However, the test results are greatly affected by the proportion of tumors and are often not comprehensive or accurate if the number of tissue samples is small. Many studies have confirmed that magnetic resonance imaging (MRI) features can reflect tumor microenvironment changes and their relationship with various genes [15,16,17], but artificially collected features may not be sufficiently comprehensive and may limit the development of models. Radiomics focuses on the lesion as whole and uses a high throughput image extraction system to enhance data collection [18]. It utilizes visual information—tumor shape, size, and grayscale—and deep information mining, such as advanced features behind tumors, to predict molecular markers noninvasively and accurately [19,20,21,22,23]. Owing to high examination costs and the invasiveness of preoperative biopsies, the detection of *CDKN2A/B* is limited in clinical practice. Hence, the preoperative, noninvasive MRI prediction of *CDKN2A/B* is of great value, as there are currently no clinically validated imaging-based markers for predicting *CDKN2A/B* homozygous deletion in gliomas. Therefore, our study creates a precedent for the preoperative noninvasive prediction of clinical *CDKN2A/B*.

This study aimed to investigate the association between conventional MRI and *CDKN2A/B,* and to establish a pre-surgery model of *CDKN2A/B* homozygous deletion in patients with gliomas. Additionally, we hoped to establish a survival analysis model for *IDH*-mutant LGGs with radiomic features combined with clinical characteristics, including *CDKN2A/B* homozygous deletion.

## 2. Materials and Methods

### 2.1. Study Population

This retrospective study included 566 patients from 2 different cohorts: the public database platform for cancer imaging—The Cancer Imaging Archive (TCIA)—with a search deadline in March 2021, and the First Affiliated Hospital of Army Medical University from January to September 2019. The need for informed consent from all patients in both cohorts was waived, and this study passed the ethics review of the First Affiliated Hospital to the Army Medical University. The inclusion criteria were as follows: patients with (1) pathologically confirmed gliomas, (2) conventional MRI sequences (fluid-attenuated inversion recovery [FLAIR], gadolinium-enhanced T1-weighted [Gd-T1W] MR), (3) *CDKN2A/B* homozygous deletion testing, and (4) general clinical information (age, sex, WHO Central Nervous System tumor classification, and histological category). The exclusion criteria were as follows: (1) lack of preoperative MR images in the postoperative MRI scan, (2) lack of either sequence in FLAIR or Gd-T1W MR, (3) presence of MRI artifacts, and (4) incomplete clinical information. Based on these criteria, 292 patients were enrolled in the first step of this study, in which imaging analysis of *CDKN2A/B* homozygous deletion was performed. In the following phase of this study, 104 patients with low-grade *IDH*-mutant gliomas were screened from the 292 enrolled patients. Clinical information such as survival status and survival period were collected for prognosis and survival analysis. For both objectives, we adapted a 7:3 random partition to divide the data into the training and validation sets. The workflow of this study is shown in (Figure 1).

### 2.2. CDKN2A/B Pathological Testing

*CDKN2A/B* was assessed using fluorescence in situ hybridization (FISH). Paraffin samples were repaired, digested, denatured, hybridized, and rinsed using FISH in our hospital. The hybridization probe used was *CDKN2A* (9p21) (Anbiping Genetic Testing Co., Ltd., Guangzhou, China [product number: F.01265, batch number: 202111001]), with the red and green fluorescent probes labeled *CDKN2A* and chromosome 9, respectively.

### 2.3. MRI Acquisition

MR images from our hospital were obtained using four different Siemens Healthcare systems.

(1)Trio TIM: 3T-MR and Gd-T1W were obtained at repetition times of 180–459 ms (echo time: 3 ms, section thickness: 2.0–5.0 mm). The repetition time for FLAIR was 7000 ms (echo time: 79 ms, section thickness: 5.0 mm).(2)Spectra: 3T-MR and Gd-T1W were obtained at repetition times of 300–400 ms (echo time: 3–4 ms, section thickness: 2.0–5.0 mm). The repetition time for FLAIR was 9000 ms (echo time: 81 ms, section thickness: 5.0 mm).(3)Avanto: 1.5T-MR and Gd-T1W were obtained at a repetition time of 204 ms (echo time: 5 ms, section thickness: 5.0 mm). The repetition time of FLAIR was 8460 ms (echo time: 134–136 ms, section thickness: 5.0 mm).(4)MAGNETOM_ESSENZA: 1.5T-MR and Gd-T1W were obtained at a repetition time of 250 ms (echo time: 5 ms, section thickness: 5.0 mm). The repetition time for FLAIR was 8000 ms (echo time: 84 ms, section thickness: 5.0 mm).

### 2.4. Post-Acquisition Preprocessing and Segmentation

Gd-T1W and FLAIR were registered for each image using a 3D Slicer (https://www.slicer.org/, accessed on 1 July 2021) to ensure that the two sequences had the same spacing and slice thickness with correction, registration, and resampling. FLAIR and Gd-T1W were selected for delineation as Gd-T1W was used to capture tumor necrosis cores and enhanced areas; FLAIR was used to observe the whole tumors and edematous areas. All MR images were normalized to reduce the effects of imaging protocol heterogeneity and apply N4-bias field correction to correct for image bias domains. Each glioma was defined as two sub-compartments of the tumor habitat: (1) the tumor parenchymal area and (2) peritumoral edema. To fully display the lesions, we first delineated the edematous region (including the inner tumor parenchyma) on the FLAIR and then delineated the inner tumor parenchyma on the Gd-T1W. A sketch of this process is shown in (Figure 2). Two radio-neurologists, blinded to the pathological results, outlined all the layers of the regions of interest (ROIs), thus forming the volumes of interest. If the two doctors had differing outlines, they discussed and summarized the final image segmentation results. The original segmentation results from the two doctors were used for consistency analysis, whereas the final results reached after discussion were used for feature extraction and radiomics analysis.

### 2.5. Radiomic Feature Extraction and Radiomic Signature Construction

The intraclass correlation consistency test examined the segmentation results of the two radiologists. The results showed that the intraclass correlation value of FLAIR was 0.68–1.00, and Gd-T1W was 0.75–1.00. For each sub-compartment (edema, tumor) in every MRI protocol, 1688 radiomic features were extracted using PyRadiomics (version 2.2.0). In total, the radiomic features were divided into four categories. The first category is first-order features, which describe simple statistics based on a gray histogram of the entire ROI, including gray mean, maximum, and minimum. The second category is the shape, which describes the geometric characteristics of the ROIs. Simple shape features include diameter and axis, whereas complex shape features include compactness and sphericity based on surface and volume algorithms. The third category includes texture features, such as the gray level cooccurrence matrix, gray level run length matrix, and gray level size zone matrix. The fourth category is higher-order statistical features, which mainly included first-order statistics and texture features derived from the wavelet transformation of the original images: exponential, square, square root, logarithm, and wavelet (wavelet-LHL, wavelet-LHH, wavelet-HLL, wavelet-LLH, wavelet-HLH, wavelet-HHH, wavelet-HHL, and wavelet-LLL).

For the first objective of building the *CDKN2A/B* predictive model, three methods of variance threshold, univariate selection, and least absolute shrinkage and selection operator (LASSO) analysis were used to select the most useful features for subsequent analysis to reduce the dimensionality of the extracted features and the redundancy of high-dimensional features (Appendix A). The LASSO Cox algorithm was used to screen the most significant radiomic features in our second objective of building the survival analysis model (Appendix A). The radiomic score (rad-score) of each lesion was calculated using a linear combination of selected features, which were weighted by their respective coefficients.

### 2.6. Clinical and Radiological Factors for Predicting CDKN2A/B

To assess the MRI features of gliomas with *CDKN2A/B* homozygous deletion, we included all patients with gliomas from the two cohorts. General clinical data (age, sex, WHO grade, and histological diagnosis) were collected. Based on the selected radiomic features, a logistic regression model was used to predict *CDKN2A/B* homozygous deletion in the training set, and a rad-score was established. A multivariate logistic regression model was established by combining the screened significant imaging features with the general clinical features to predict *CDKN2A/B* homozygous deletion.

### 2.7. Prognostic Value Analysis

We screened 104 patients to evaluate the effect of *CDKN2A/B* homozygous deletion and other clinical characteristics on the prognosis of patients with *IDH*-mutant LGGs. To investigate the association between clinical characteristics and overall survival (OS), we used a univariate Cox regression to screen out clinical features related to prognosis; we combined these relevant clinical features with radiomic features to construct a prognostic model of *IDH*-mutant LGGs using a multivariate Cox regression analysis.

For each MRI protocol, each patient in the training and validation set was assigned all risk scores in the high- and low-risk score groups based on the rad-score median point. The prognostic power of clinical and imaging features was assessed using the Kaplan–Meier (K–M) survival analysis.

The K–M survival curve was used to compare the survival period of the high- and low-risk groups. The horizontal and vertical axes on the K–M survival curve represent time and survival probability, respectively. At any given point on the survival curve, the probability that a patient in each group is still alive at that time is shown. In addition, the hazard ratio was used to quantify the effect of individual characteristics on survival. The predictive performance of the model was quantified using the C-index. Adjustments were made based on the agreement between the estimated 1-, 3-, and 5-year survival rates and the actual year survival rates in the training and validation cohorts with corresponding calibration curves. *p* < 0.05 was considered statistically significant.

### 2.8. Statistical Analysis

Normalization of features, selection of features, and model construction were performed using Python (version 3.7.0), Scikit-learn version 0.19.2, and PyRadiomics (version 2.2.0). Other statistical analyses were performed using the R software (version 3.3.0). Statistical significance was set at *p* < 0.05.

For the first objective of predicting *CDKN2A/B*, univariate analysis was performed to select candidate features with significant differences (*p* < 0.05). The best subset selection was used for the subsequent model construction. The chi-squared test or Fisher’s exact test and the Mann–Whitney U test were used for categorical and continuous variables, respectively. The receiver operating characteristic (ROC) curve analysis was used to calculate the area under the curve (AUC), precision, sensitivity, and specificity to evaluate the predictive effect of the model.

For the second objective of the survival analysis, the survival probability was analyzed using the K–M method. The log-rank test was used to compare outcomes between patients with high- and low-risk scores. Univariate Cox regression analyses were performed to identify the significant predictors of survival probability.

## 3. Results

### 3.1. Clinical Characteristics

Pre-modeling analysis of all clinical features revealed that only age, histological classification, and WHO grade were statistically significant (*p* < 0.05) between the two groups. These factors were associated with *CDKN2A/B* homozygous deletion (Table 1).

### 3.2. CDKN2A/B Predictive Model Construction and Evaluation

The selected radiomic features were used to establish a univariate logistic regression model, and the three statistically significant clinical factors were used to establish a clinical univariate model. A comprehensive model combining radiomic features with clinical factors was then constructed using a multiple logistic regression analysis. The results are shown in (Table 2).

The AUCs of the comprehensive model in the training and validation sets were 0.880 (95% confidence interval [CI], 0.840–0.918) and 0.825 (95% CI, 0.743–0.894), respectively. The results of the multivariate logistic regression model are presented as nomograms (Figure 3). The results showed that the rad-score was strongly associated with *CDKN2A/B* homozygous deletion. It also showed that the radiomics model based on the conventional MRI sequence had a better predictive effect on *CDKN2A/B* homozygous deletion.

Calibration curve analysis was performed on the model, and the results of the calibration curve reflected the proximity of the predicted results of the model and the actual results. In our study, the calibration curve reflected a high degree of agreement between the predictions of the model in the training and validation sets and the actual status, indicating that the model displayed good performance and reliability in predicting *CDKN2A/B* homozygous deletion in gliomas (Figure 4).

### 3.3. Prognostic Value of the Fusion Radiomics Signature and Clinical Characteristics

A total of 1688 features were extracted from every conventional sequence, FLAIR, and Gd-T1W. The 23 most significant image features were extracted.

Univariate Cox analysis of all clinical features showed that only the *CDKN2A/B* homozygous deletion status was statistically significant (*p* < 0.05) (Appendix A). This result was expected as it agrees with the previous studies demonstrating that *CDKN2A/B* homozygous deletion is an independent prognostic marker in LGGs.

By combining *CDKN2A/B* homozygosity deletion with radiomic features, a multivariate Cox regression survival analysis model was established. The C-indices in the training and validation sets were 0.876 and 0.818, respectively (Appendix A). The model was displayed using a nomogram (Figure 5).

Next, all risk scores were divided into high- and low-risk scores according to the median value of patient risk scores. K–M survival analysis was used to perform univariate survival analysis according to the high- and low-risk scores. The results showed that high- and low-risk scores were the key influencing factors for patient survival (*p* < 0.05). The results of the K–M growth curve showed significant differences between the high- and low-risk groups (Figure 6).

In clinical practice, patient-individualized treatment plans can be guided by the patient’s risk score. The calibration curves indicated that the predicted and actual survival outcomes were good at 2, 4, and 6 years of follow-up in the training and validation sets (Appendix A).

## 4. Discussion

In this study, we investigated a noninvasive imaging method based on conventional MRI to establish radiomics models for predicting *CDKN2A/B* homozygosity deletion in gliomas, and the results showed that the logistic regression integrated model had the most excellent predictive performance (AUC of 0.880 and 0.825 for the training and validation sets, respectively). This study is the first to propose the use of noninvasive imaging to build radiomic models for *CDKN2A/B* homozygosity deletion in gliomas. *CDKN2A/B* homozygosity deletion serves as a prognostic predictor for gliomas, predicting poorer OS, and providing treatment guidance. Therefore, accurately predicting this subtype provides an important clinical adjunct value for accurate diagnosis, clinical decision making, and survival prediction for molecular staging of patients with gliomas.

First, we adopted logistic regression models to analyze the accuracy and reliability of independent imaging and clinical features, and then combined predictions from both. The results showed that combining radiomic features with clinical parameters (age and sex) did not greatly improve the models’ performance, which fully characterized the importance and accuracy of image characteristics in model construction. Previous studies have reported the correlation of MRI features with *CDKN2A/B* homozygosity deletion. This study expanded on this research and used the newest machine learning algorithms to increase the reliability of gene prediction models.

Since few studies have used MRI features to explore *CDKN2A/B* homozygous deletion, the Heidelberg University Medical Center analyzed the association between multiparametric, multiregional MRI features and key molecular features of GBM, including *CDKN2A*, and found that *CDKN2A* deletion is associated with increased normalized relative cerebral blood volume (nrCBV) and normalized regional cerebral blood flow (nrCBF) within the enhanced tumor volume (ROC range, 63% to 69%; false discovery rate adjusted *p* < 0.05) [24]. This study found meaningful associations between the multiparametric MRI features and molecular markers. Since it was a multi-genotype study in GBM, it did not investigate the relevant situation in LGGs; however, *CDKN2A/B* homozygous deletion is implicated in LGGs. Our study comprehensively constructed a *CDKN2A/B* predictive model based on MRI features for all patients with gliomas and overcame aforementioned shortcomings. Concurrently, Nowak et al. identified significantly different imaging parameters in supratentorial ependymomas according to *CDKN2A* alterations [25]. Since the stratification of patients by morphological MRI alone is difficult, radiomic technology promises to make such research models more accurate and reliable. Our study innovatively used radiomics to determine the association between MRI features and *CDKN2A/B* homozygous deletion in gliomas, constructing a *CDKN2A/B* predictive model. Through this study, we aimed to construct a template for *CDKN2A/B* homozygous deletion study models for gliomas, contributing to the development of clinical treatment plans.

Second, we investigated the association between OS and all clinical features, including *CDKN2A/B* homozygous deletion, in *IDH*-mutant LGGs. We found that only *CDKN2A/B*, an independent prognostic factor for *IDH*-mutant LGGs, was associated with prognoses. According to the rad-score median point (−0.07), patients were divided into high- and low-risk groups. Patients in the high-risk group were those with rad-scores greater than −0.07. The K–M curve showed that patients in the high-risk group had different degrees of death over time, and the risk of death was higher than in the low-risk group. However, no patients in the low-risk group died during the follow-up period. This study reveals that *IDH*-mutant LGGs have an overall optimistic prognosis, but certain patients have a poorer prognosis, such as those with *CDKN2A/B* homozygous deletion. Radiomics features can help identify which class of patients is at risk. This verifies that in our dataset, patients with *CDKN2A/B* homozygous deletion and a rad-score of more than −0.07 have a poor prognosis, and radiomic features can be used as noninvasive prognostic markers for patients with LGGs.

The latest CNS tumor classification has indicated that *CDKN2A/B* homozygous deletion represents a poor prognosis in astrocytoma [14], but there are no reports using *CDKN2A/B* and MRI features to predict survival in LGGs. This study is novel in that *CDKN2A/B* and conventional MRI features can be used to predict the survival of patients with preoperative LGGs, providing a reference for clinical prognosis. In addition, this study aimed to predict survival in subgroups of LGGs, including astrocytoma, oligodendroglioma, WHO grade, and whether postoperative adjuvant radiotherapy was required, to determine OS in these LGG subgroups and whether patients would benefit from postoperative adjuvant radiotherapy. However, when we analyzed the clinical characteristics, these indicators were not statistically significant, and only *CDKN2A/B* homozygous deletion was included in the subsequent analysis. This limited the subgroup analysis, which may have been related to the enrolled patients. We plan to expand the sample size in future studies.

This study has some limitations. First, the data from TCIA and our hospital do not coincide on the MRI scan protocol; therefore, to keep all MR images in a consistent grayscale, we excluded most of the patients from TCIA. Second, although the expert consensus on glioma segmentation has been published, some gliomas—particularly LGGs—have unclear demarcations between tumor parenchyma and edema which makes it difficult to separate the necrotic components in the center from the surrounding strengthened tissue. This is also why we decided to divide the tumor into two parts. Moreover, the segmentation is time-consuming, and the results will vary due to different delineators and times, which makes the reproducibility of radiomics poor. In the future, the development of fully automatic delineation tools and accurate segmentation will be helpful for clinical model reconstruction. However, the validation was retrospective, and the LGG sample size in the survival analysis was small. We plan to include a large-scale multi-site evaluation in further studies. In the future, our study will also focus on incorporating complementary imaging parameters (PET, perfusion, diffusion-weighted imaging), which may further improve survival prediction using radiogenic analysis while also considering subsequent treatments.

## 5. Conclusions

We were able to successfully develop superior *CDKN2A/B* homozygous deletion predictive and *IDH*-mutant LGG survival models. We found that MRI can be used as a noninvasive imaging-based marker for detecting genes and prognostic indicators to guide clinical treatment decisions. Radiomics displays high accuracy and reliability in predicting *CDKN2A/B* homozygous deletion, which provides a stepping stone for future clinical applications of molecular genetic diagnostic platforms or AI software in diagnosing gliomas.

## Figures and Tables

**Figure 1 brainsci-13-00548-f001:**
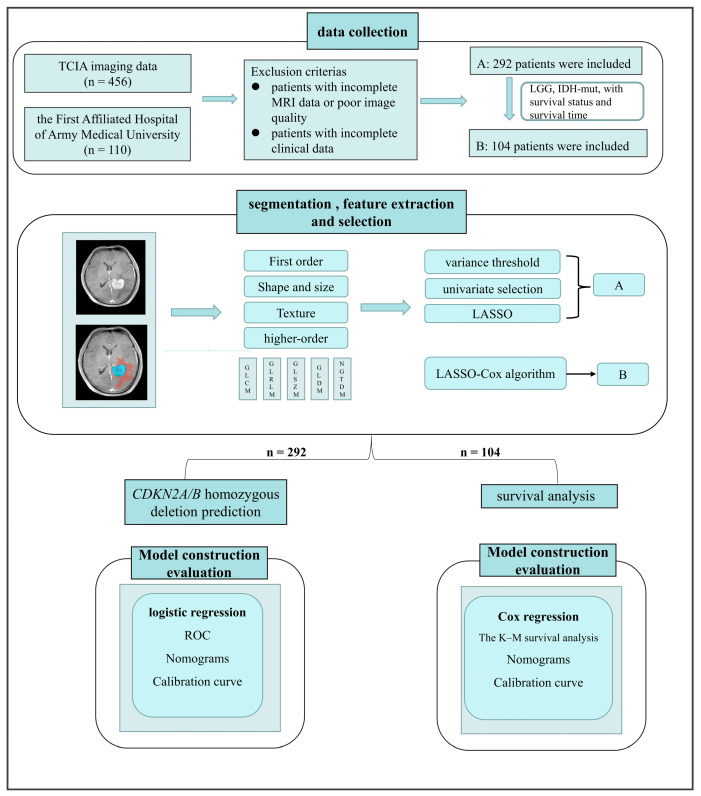
Workflow of this study: A means the project of *CDKN2A/B* homozygous deletion prediction. B means the project of survival analysis.

**Figure 2 brainsci-13-00548-f002:**
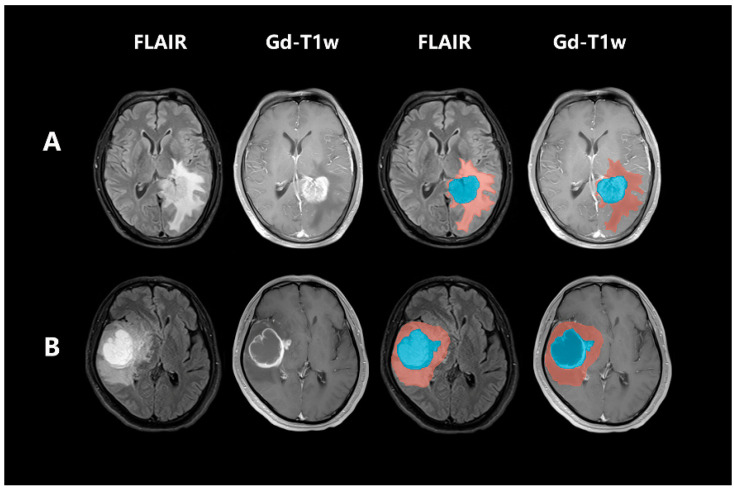
The delineated and displayed regions of interest. Blue indicates the tumor parenchyma and orange indicates the edematous region. (**A**) Patients without *CDKN2A/B* homozygous deletion and (**B**) Patients with *CDKN2A/B* homozygous deletion.

**Figure 3 brainsci-13-00548-f003:**
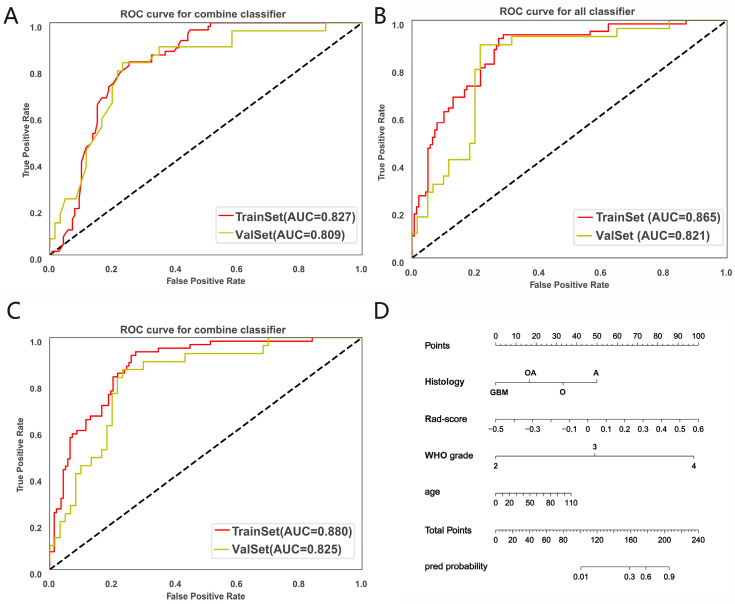
Establishment and validation of the predictive model of *CDKN2A/B* homozygous deletion in glioma. (**A**–**C**): The receiver operating characteristic results of the models. (**A**) clinical model, (**B**) radiomics model, (**C**) the combined clinical and radiomics comprehensive prediction model. (**D**) Nomogram of the combined prediction model of *CDKN2A/B* homozygous deletion: the relevant clinical characteristics and rad-score were included in the nomogram, and the results showed that rad-score correlated most strongly with the model.

**Figure 4 brainsci-13-00548-f004:**
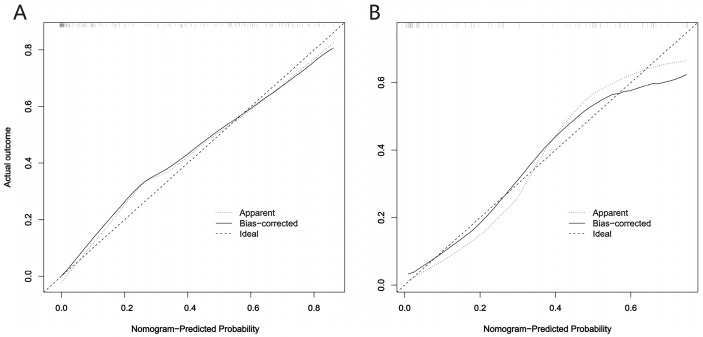
Calibration curves for the prediction model of *CDKN2A/B* homozygous deletions. (**A**) Training set and (**B**) Validation set.

**Figure 5 brainsci-13-00548-f005:**
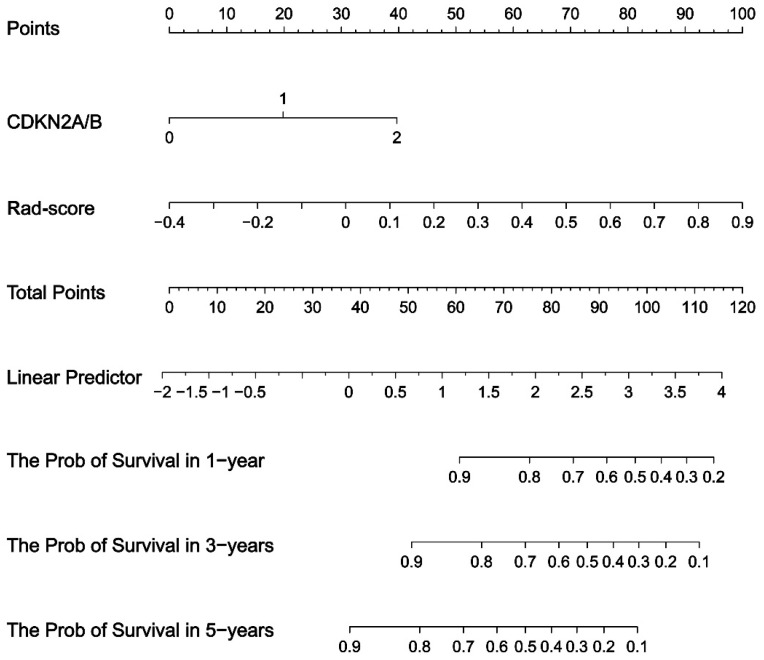
Nomogram of the training set of survival analysis: survival rate at 1-, 3-, and 5-years can be calculated from *CDKN2A/B* homozygous deletion status and rad-score.

**Figure 6 brainsci-13-00548-f006:**
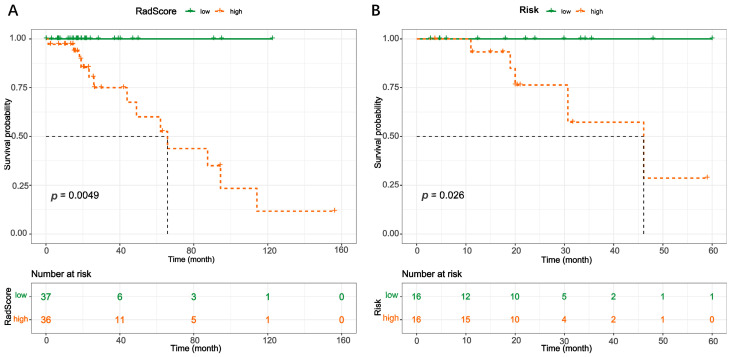
Kaplan–Meier curves with isocitrate dehydrogenase-mutant low-grade glioma. (**A**) Training set and (**B**) Validation set. Mortality was higher in the high-risk group, whereas almost no patients died in the low-risk group (*p*-value was 0.026 in the training group and 0.0049 in the validation group). The number at risk means the number of people in the high and low groups who were not censored from the endpoint event at different times.

**Table 1 brainsci-13-00548-t001:** The baseline characteristics of the patients.

		*CDKN2A/B* Homozygous Deletion			
		NO	YES	*p*-Value	SMD
N		198	94		
**Age (median [IQR])**		41.50 [24.00, 57.75]	60.00 [52.25, 66.00]	<0.001	0.856
Gender (%)	Female	111 (56.1)	61 (64.9)	0.192	0.181
	Male	87 (43.9)	33 (35.1)		
**Histology**	Astrocytoma	43 (21.7)	8 (8.5)	<0.001	1.249
	Oligoastrocytoma	27 (13.6)	2 (2.1)		
	Oligodendroglioma	51 (25.8)	1 (1.1)	.	
	GBM	77 (38.9)	83 (88.3)		
**WHO grade (%)**	Ⅱ	65 (32.8)	1 (1.1)	<0.001	1.262
	Ⅲ	56 (28.3)	10 (10.6)		
	Ⅳ	77 (38.9)	83 (88.3)		

The bold means these clinical characteristics were statistically significant (*p* < 0.05).

**Table 2 brainsci-13-00548-t002:** Model results for prediction of *CDKN2A/B* homozygous deletion.

Dataset	Model	AUC	95%CI	ACC	Sensitivity	Specificity
Training set	radomics	0.865	0.820~0.909	0.793	0.631	0.87
	clinic	0.824	0.776~0.871	0.783	0.723	0.812
	radiomic + clinic	0.88	0.840~0.918	0.793	0.662	0.855
Validation set	radiomic	0.821	0.741~0.890	0.708	0.483	0.817
	clinic	0.802	0.722~0.875	0.753	0.655	0.8
	radiomic + clinic	0.825	0.743~0.894	0.73	0.517	0.833

Radomics: Model based on radiomic features only. Clinic: Model based on clinical characteristics only. Radiomic + Clinic: Model based on radiomic features and clinical characteristics.

## Data Availability

The datasets used and/or analyzed during the current study are available from the corresponding author on reasonable request.

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
