# Peer review of "Cyclin-Dependent Kinase Inhibitor 2A/B Homozygous Deletion Prediction and Survival Analysis"

_brainsci, 2023, doi:10.3390/brainsci13040548_

Round 1

Reviewer 1 Report (New Reviewer)

The authors aimed to investigate the association between conventional MRI and CDKN2A/B and to establish a model of CDKN2A/B homozygous deletion in patients with glioma before surgery. Additionally, we hoped to establish a survival analysis model for IDH-mutant LGGs with radiomic features combined with clinical characteristics, including CDKN2A/B homozygous deletion. Albeit, I consider these findings to provide new insight into cancer-related fields, I still have some suggestions.

1, Most figures and tables are highly professional; however, the authors should guide the readers to the meaning of the images and tables appropriately; otherwise, it is likely to cause misunderstandings. Therefore, I suggest the author consider revising these figures and table legends again.

2, The author developed superior CDKN2A/B homozygous deletion predictive and IDH-mutant LGG survival models. However, It would be much better if the authors could provide some Workflow or Scheme for this research, I suggest that they can take a look at the recent paper in MDPI (PMID: 35328243, 24619302, 34834441)

3, Figure 3-4 suggests that CDKN2A/B homozygous deletion is an independent prognostic marker in LGG. However, It would be much better if the authors can validate their data via proteinatlas, and discuss these methodologies and literature as well as the validated data for cancer recurrence or metastasis in the manuscript (PMID: 22588877, 25613900, 32064155)

4, There are few typo issues for the authors to pay attention; please also unify the writing of scientific terms. “Italic, capital”?  

5, The font is too small for some of the current figures; meanwhile, the manuscript also needs English proofreading.

Author Response

Reviewer 2 Report (New Reviewer)

In this manuscript, Yang et al. developed CDKN2A/B homozygous deletion predictive and IDH-mutant LGG survival models by using MRI as a noninvasive technique. This study is remarkable for prognosis and treatment of glioma through predicting CDKN2A/B homozygous deletion by utlizing MRI features, otherwise a very high cost application. Manuscript is well-written and has a clear written expression. Despite the shortcomings of the study which they stated in the results part, it is still promising to give hope for accurate diagnosis and targeted therapy for glioma.

Author Response

Thank you for your comments. We will continue to improve the article and strive to do better.

This manuscript is a resubmission of an earlier submission. The following is a list of the peer review reports and author responses from that submission.

Round 1

Reviewer 1 Report

Dera Editor,

I reviewed the manuscript by Yang et al., entitled “Cyclin-Dependent Kinase Inhibitor 2A/B Homozygous Deletion Prediction and Survival Analysis”. The aim of this study is to investigate the association between conventional MRI and CDKN2A/B and CDKN2A/B homozygous deletion predictive marker .in patients with glioma before surgery.  The manuscript is well written , the results are well described and  well presented. This study may provide a new information for the reader of the journal. Accordingly, the study can be published in the present form.

Many thanks